# Influence of the Type and Use of Soil on the Distribution of Organic Carbon and Other Soil Properties in a Sustainable and Resilient Agropolitan System

Pura Marín-Sanleandro [1],*, Ana María Gómez-García [1],†, Arantzazu Blanco-Bernardeau [2], Juana María Gil-Vázquez [1] and María Asunción Alías-Linares [1]

[1] Department of Agricultural Chemistry, Geology and Pedology, Faculty of Chemistry, University of Murcia, Campus de Espinardo, 30100 Murcia, Spain; farmaciagomez@gmail.com (A.M.G.-G.)
[2] KVeloce I+D+i, Senior Europa SL, Plaza de la Reina 9, Esc A 1°B, 46002 Valencia, Spain; ablanco@kveloce.com
* Correspondence: pumasan@um.es; Tel.: +34-659634777
† This work was part of the doctoral thesis of the second author Ana María Gómez García. Doctoral program in Universidad de Murcia, 30100 Murcia, Spain.

**Abstract:** Urban and peri-urban agriculture is one of the strategies that emerged on the path towards agri-food sustainability in cities. This paper aims at improving the knowledge of the soil properties in a peri-urban area and their agricultural potential to support ecosystems with biodiversity worth conserving. The study area was located in the mid-plain of the Segura River (SE Spain). Arable soil layer samples were collected at 68 points to assess the distribution of organic carbon and to study other indicators of soil quality. A Wilcoxon and Kruskal–Wallis test was conducted to compare between the types of soils present in the area (calcaric Fluvisols and calcaric Regosols) and soil uses (industrial, cultivated and abandoned). Statistical analysis indicates that there are significant differences (at the 0.05 significance level) between Fluvisols and Regosols ($p$ value = $3.65 \times 10^{-5}$). Regarding use, the abandoned Fluvisols presented an average value of 9.33 g kg$^{-1}$ of OC while the Fluvisols that are currently cultivated have a higher average content of OC (11.35 g kg$^{-1}$). For soils under industrial use, the average OC content is 5.13 g kg$^{-1}$. Spatial distribution of organic carbon in these soils depends on the type of soil and the human influence on them, including the use.

**Keywords:** sustainable development; agropolis; agricultural soils; soil uses; resilient system; soil organic carbon; Fluvisols; Regosols; ecosystem services; peri-urban system

## 1. Introduction

The challenge that world society, and more particularly the Region of Murcia, faces in the coming years is to achieve developed, sustainable and resilient agropolitan systems [1]. We must produce healthy food for an urban population that does not stop growing, with increasingly reduced physical space to cultivate and scarcity of water, particularly in arid areas such as the Region of Murcia, and all this in a panorama of climate change and economic and social uncertainty aggravated by the global COVID-19 pandemic and the war in Ukraine.

Urban and peri-urban agriculture are strategies that emerged on the path towards agri-food sustainability in cities. These are understood as agricultural practices that are carried out, regardless of legal situation, in and around the city, in public or private spaces, and by an individual or collective initiative of citizens and/or facilitated by public organizations or private companies that perform multiple functions [2,3]. Very old sustainable agropolitan systems [4] and current ones are present in many parts of the world where peri-urban areas with several plans have promoted state policies [5].

Historically, ecosystem services were used, being exploited and enjoyed. Urban populations still rely on ecosystems, but prioritize non-ecosystem services, particularly

socioeconomic benefits. Population growth and densification increase the scale and change the nature of both ecosystem- and non-ecosystem-service supply and demand, weakening direct feedback between ecosystems and societies and potentially pushing socioecological systems into traps that can lead to collapse. The interacting and mutually reinforcing processes of technological change, population growth and urbanization contribute to the exploitation of ecosystems through complex relationships that have important implications for sustainable resource use [6]. The potential of peri-urban agrarian ecosystems is recognized as one of the cornerstones for improving urban sustainability [7]. Furthermore, systems referred as urban agroforestry (UAF), combining trees with other crops or with the trees themselves as crops (such as fruit trees), may offer greater cultural and ecological benefits [8,9] through their potential to infiltrate stormwater, mitigate heat island effects, sequester carbon, and contribute to soil formation, among other effects [8,10,11].

The direct benefits of urban and peri-urban agriculture have been widely studied (i.e., [12–14]), to which its enormous potential to mitigate the effects of climate change, reduce energy consumption, and contribute to environmental and human health in cities must be added [15,16], as well as for enhancing the social integration of the poorest. In this sense, urban planning [17] must articulate the productive, ecological, landscape and urban functions of peri-urban agricultural spaces, establishing a gradient between rural and urban areas.

Despite the mutation suffered in recent decades due to urban pressure, the "Huerta de Murcia" (Murcia traditional irrigated system) is a space of great agroecological, economic, social and environmental interest [12,14]. A process of agroecological revitalization has to be set in motion to make the Huerta a developed and sustainable system. The Huerta is considered an ecosystem service, and these are directly influenced by changes in land use resulting from ecological restoration and urbanization, globally or locally [3,18].

In these agropolitan systems, soil organic carbon (OC) is fundamental as an important component of the global C cycle, representing 69.8% of the carbon in the biosphere [19]. The soil can act as a source or reservoir of C depending on its use and management [20,21]. The loss of humic material from cultivated soils is higher than the humus formation rate in undisturbed soils. Therefore, the soil under conventional cultivation conditions is a source of $CO_2$ for the atmosphere [22]. In these agropolitan systems, there are agronomic practices that may enhance the capture of C in the soil [23,24]. Conservation tillage [25], which includes zero tillage [19], is a soil management system that has a high potential capacity to sequester carbon in the soil, fostering the adaptation of agricultural practices to climate change [26].

In the last few years, there has been an increase in soil organic carbon studies concerning climate change and the possible role of the soil as a carbon sink or store [27–29]. Soil organic carbon affects most of the chemical, physical, and biological properties of the soil related to its quality, sustainability, and productive capacity [30]. Thus, organic carbon must be maintained or increased in sustainable management systems as an essential requirement towards resilience. However, compared with natural and agricultural ecosystems, few studies focus on SOC sequestration in urban ecosystems [31].

In the mid-plain of the Segura River, the rural landscape has been transformed by urbanization and industrialization. The irrigated landscape has diminished due to the expansion of urban nuclei, the sparse growth in medium-density residential areas and the creation of a kind of continuum of spaces where industrial districts follow one another. Thus, it is a rural landscape transformed by urbanization and industrialization.

The overall objective of this study is to improve the knowledge of the agricultural potential of these soils, based mainly on their content in organic carbon, to redirect the change of uses and to discover which ones support ecosystems with biodiversity worth conserving. This will contribute to rational planning of the territory, pointing to each land area's most appropriate use according to its intrinsic and extrinsic characteristics, to contribute towards a sustainable and resilient agropolitan system. Environmental planning and territorial planning and management intend, ultimately, to resolve conflicts between

the different interests and social demands that arise regarding the use and conservation of space and its resources [32]. To this aim, knowing the soil properties represents an essential step to support and preserve their agroecological potential. The purpose of this paper is to perform a development forecast according to the characteristics of the soils and their uses, as previously carried out in other geographical areas (i.e., [33]).

This study has been deemed necessary to respond to the need for detailed, current and reliable information on the characteristics of the soils in the Vega del Segura (middle plain of Segura River, Murcia, Spain) and to relate them to their types and uses, as well as the possible overexploitation to which water and soil may be subject. The unavailability of previous soil and geoenvironmental studies in the area fully justifies it.

The starting hypothesis is that the studied soils will be given agricultural or industrial use according to their intrinsic characteristics—preferably, Regosols will be in industrial use and Fluvisols under agricultural use.

## 2. Materials and Methods

### 2.1. Characteristics of the Study Area

The physical environment is characterized by the presence of two unique relief units: the Segura River valley, which constitutes an alluvial depression, and the foothills of the Betic mountain ranges—small, low-lying hills. The main watercourse in the study area is the Segura River, the third-largest Spanish Mediterranean river, with a predominantly north-northwest–southeast direction, which appears embedded between limestone reliefs, gradually opening up and constituting a small floodplain. In addition, the study area is included in the aquifer system that covers the entire Segura mid-plain. The prolonged dry season is aggravated by the frequent intakes of water that are carried out and conducted along the different ditches to irrigate the plains. The climatic characteristics (average annual precipitation of 320 mm and average annual temperature of 17 °C) establish a Mediterranean regime with a marked semiarid character, with the coincidence of maximum temperatures with minimum rainfall. The soil moisture regime is aridic, but it is considered xeric in the vicinity of the river and in irrigated soils, which is related to the lateral water dynamics from the river and the traditional irrigation system (by gravity) consisting of periodic land flooding. In addition, several plant ecosystems have been recognized, generally represented by stages of high degradation, among them a thermo-Mediterranean and a meso-Mediterranean series, as well as the presence of the Ibero-Levantine riparian and Murcian halophilous macroseries.

Two main types of soils can be found in the area: calcaric Fluvisols and calcaric Regosols [34]. Fluvisols are scarcely evolved soils with a great agricultural vocation, developed from quaternary materials deposited by the river, with a simple Ap-C type profile, in which the C horizon can present different sub-horizons and even constitute lithological discontinuities of each contribution of the river in different avenues. They are called calcaric because they have a lot of calcium carbonate. Calcaric Regosols are young soils, with a simple A-C type profile, developed from Miocene marls that in many cases have an abundance of gypsum, even visible in crystals in the field.

The study area includes areas with different uses, the most significant being agricultural and industrial. Urban areas such as Molina de Segura, Alguazas, Torres de Cotillas, Ceutí and Lorquí, where the average population density is 400 inhabitants/km$^2$, are included. Almost the entire territory is dedicated to irrigated crops, among which citrus, orange and lemon trees, and stone fruit trees predominate. Horticultural and herbaceous crops are very common and are intended more for family consumption. In the study area, there are several industrial estates derived from the needs of fruit and vegetable production in the Vega.

### 2.2. Sampling

The area selected for this work was the central sector of the mid-plain (Vega Media) of the Segura River (Murcia), with an approximate extension of 65 km$^2$ (Figure 1). A sampling

of the arable layer (0–30 cm) of the soils was carried out following a $1 \times 1$ km$^2$ grid, which represented a total of 68 samples, whose situation is given in Table S1 (Figure 2). In turn, each sample is a mixture of three subsamples to make it homogeneous.

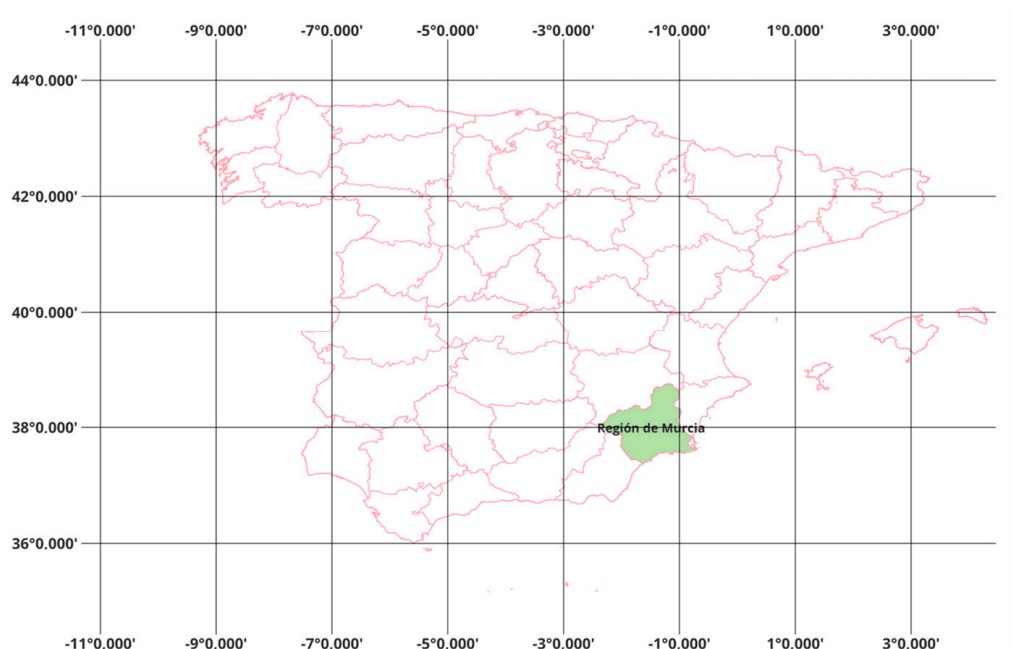

**Figure 1.** Location of the study area.

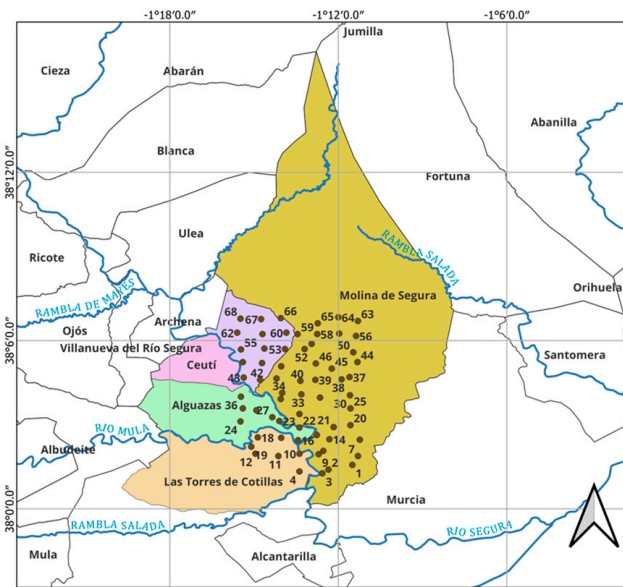

**Figure 2.** Study area and sample location.

In the case of the Fluvisols, 20 of them were cultivated, while 14 were not cultivated at the time, but had been cultivated in the past. In the case of the Regosols (33 samples), their use is industrial, since they are located in five large industrial estates, with industries predominantly from the food sector (especially traditional canning), but mixed with other sectors. A haplic solonchak was also sampled (sample 48), but was not considered representative and therefore not included in the results and discussion. The values of each of the 68 samples studied are presented in Tables S1–S5 of the Supplementary Material.

### 2.3. Soil Analyses

Once the samples were taken, they were air-dried, sieved to 2 mm and ground after removing fresh organic remains. The necessary analytical determinations have been made for an adequate typological characterization of the soils, according to the IUSS Working Group WRB system [34].

The particle size distribution (clay, silt and sand) of the soil samples was determined by Robinson's pipette method [35]. Particle density was calculated by the pycnometer method. The moisture content (Hum.) of the soil samples is obtained by weighing the difference once they have been dried in an oven at 105 °C.

The chemical parameters were determined as follows: organic carbon (OC) content ([36] and modified by [37]), total nitrogen (TN) by Kjeldahl's method, as described by [37], pH in a 1:5 suspension of soil in water, and electrical conductivity (EC) in a soil–water ratio of 1:5 [38]. Determination of total carbonate ($CaCO_3$) was performed by volumetric analysis using a Bernard calcimeter; cation exchange capacity (CEC); sodium, potassium and magnesium cations by atomic absorption; and phosphorus by Watanabe and Olsen's method, all as described by [35]. The available elements—iron, copper, manganese, and zinc—were determined by inductively coupled argon plasma mass spectrometry (ICP-MS model Agilent 7900) after extraction with a solution of 0.05 M DTPA, 0.01 M $CaCl_2$ and 0.1 M triethanolamine at pH 7.3 [35]. To determine the total metal concentration (Fe, Cu, Mn, and Z an acid digestion with aqua regia ($HNO_3/HCl$, 1:3) in a microwave oven at 220 °C for 1 h [39] and subsequently, inductively coupled argon plasma mass spectrometry (ICP-MS model Agilent 7900) was used.

### 2.4. Statistical Analyses and GIS

To carry out a statistical study of the results obtained from the arable layer samples, the statistical package R [40] was used to analyze whether the differences observed between the samples concerning to the variables studied were significant regarding the type and use of the land through nonparametric methods (Kruskal–Wallis and Wilcoxon), since it was not possible to ensure the normality and homoscedasticity of the analyzed variables. A two-way analysis with soil type–soil use interaction was not deemed necessary, since the use of soil corresponds to the soil type, as explained in Section 2.2. For cartographic representation, QGIS 3.22 Białowieża [41] was used, which allows the interpolation of the values by IDW (inverse weighting at distance).

### 3. Results and Discussion

In the studied area, there are two types of soils, which are found in a similar proportion: calcaric Fluvisols and calcaric Regosols, while haplic solonchaks appear occasionally in salt marshes [34]. The Fluvisols are currently under agricultural use or were dedicated to cultivation in the past and are now fallow or abandoned. Calcaric Regosols are located in industrial or urban areas.

Regarding the variables studied, Tables 1 and 2 show their mean values and standard deviation, as well as the nonparametric Kruskal–Wallis test results, carried out depending on the type of soil (calcaric Fluvisols, calcaric Regosols) and the use to which they are dedicated (industrial, current cultivation and abandoned cultivation). A nonparametric test was selected since neither the independence of the observations in the groups nor the normality of variables could be ensured.

The results of the Wilcoxon test are shown in Tables 3 and 4, carried out according to the type of soil (calcaric Fluvisol sand, calcaric Regosols) and the use to which they are put (industrial, current cultivation and abandoned cultivation).

**Table 1.** Statistics of the variables studied according to the type of soil. Kruskal–Wallis nonparametric test. silt + clay (%), Density (g/cm$^3$), Hum (%), OC (g kg$^{-1}$), TN (g kg$^{-1}$), CaCO$_3$ (g kg$^{-1}$), pH (1:5), EC (μS cm$^{-1}$), P$_{available}$ (mg/kg), CEC (mE/100 g), K$_{available}$ (mg/100 g), Mg$_{available}$ (mg/100 g), Na$_{available}$ (mg/100 g), Cu$_{available}$ (mg/kg), Fe$_{available}$ (mg/kg), Mn$_{available}$ (mg/kg), Zn$_{available}$ (mg/kg), Cu$_{total}$ (mg/kg), Fe$_{total}$ (mg/kg), Mn$_{total}$ (mg/kg), Zn$_{total}$ (mg/kg).

| Variable | Average | | SD | | Kruskal df = 2 Chi-Square | *p* Value |
|---|---|---|---|---|---|---|
| | Fluvisol $n = 34$ | Regosol $n = 33$ | Fluvisol $n = 34$ | Regosol $n = 33$ | | |
| silt + clay | 75.478 | 74.185 | 11.729 | 12.967 | 20,374 | 0.3611 |
| Density | 19.647 | 18.771 | 14.330 | 11.770 | 22,591 | 0.3232 |
| Hum. | 3.396 | 4.160 | 1.973 | 2.411 | 20,631 | 0.3565 |
| OC | 10.582 | 5.262 | 4.693 | 3.458 | 204,374 | 0.03648 *** |
| TN | 2.914 | 1.834 | 1.045 | 0.872 | 223,178 | 0.01425 *** |
| C/N | 3.801 | 3.112 | 1.601 | 1.566 | 54,942 | 0.0641 |
| CaCO$_3$ | 351.793 | 363.379 | 60.563 | 77.569 | 1606 | 0.448 |
| pH | 8.546 | 8.337 | 0.301 | 0.387 | 56,067 | 0.0606 |
| EC | 481.63 | 1133.12 | 437.12 | 665.38 | 156,689 | 0.000395 *** |
| P$_{available}$ | 23.895 | 9.518 | 13.639 | 9.861 | 29,9491 | 0.000314 *** |
| CEC | 14.697 | 13.313 | 3.655 | 3.834 | 37,002 | 0.1572 |
| K$_{available}$ | 35.621 | 25.008 | 20.153 | 12.608 | 57,954 | 0.05515 |
| Mg$_{available}$ | 47.240 | 42.007 | 23.552 | 28.650 | 23,651 | 0.3065 |
| Na$_{available}$ | 29.306 | 64.139 | 20.153 | 12.608 | 0.0915 | 0.9553 |
| Cu$_{available}$ | 6.591 | 1.598 | 5.280 | 1.6908 | 291,275 | 0.000473 *** |
| Fe$_{available}$ | 1.758 | 0.555 | 1.858 | 0.540 | 265,556 | 0.001712 *** |
| Mn$_{available}$ | 7.345 | 3.421 | 4.850 | 3.100 | 17,182 | 0.000185 *** |
| Zn$_{available}$ | 1.980 | 0.731 | 1.238 | 0.742 | 247,352 | 0.004254 *** |
| Cu$_{total}$ | 89.629 | 62.098 | 125.890 | 77.878 | 50,501 | 0.08006 |
| Fe$_{total}$ | 11599.6 | 10553.3 | 2350,32 | 2041.30 | 45,552 | 0.1025 |
| Mn$_{total}$ | 224.513 | 212.228 | 41.274 | 39.111 | 3328 | 0.1894 |
| Zn$_{total}$ | 44.160 | 23.754 | 33.412 | 21.377 | 119,264 | 0.002572 ** |

$p < 0.001$ ***; $p < 0.01$ **.

**Table 2.** Statistics of the variables studied based on land use. Kruskal–Wallis nonparametric test. Silt + clay (%), Density (g/cm$^3$), Hum (%), OC (g kg$^{-1}$), TN (g kg$^{-1}$), CaCO$_3$ (g kg$^{-1}$), pH, EC (μS cm$^{-1}$), P$_{available}$ (mg/kg), CEC (mE/100 g), K$_{available}$ (mg/100 g), Mg$_{available}$ (mg/100 g), Na$_{available}$ (mg/100 g), Cu$_{available}$ (mg/kg), Fe$_{available}$ (mg/kg), Mn$_{available}$ (mg/kg), Zn$_{available}$ (mg/kg), Cu$_{total}$ (mg/kg), Fe$_{total}$ (mg/kg), Mn$_{total}$ (mg/kg), Zn$_{total}$ (mg/kg).

| Variable | Use | | | | | | Kruskal (df = 2) | |
|---|---|---|---|---|---|---|---|---|
| | Abandoned N = 14 | | Cultivated N = 20 | | Industrial N = 33 | | Chi-Square | *p* Value $\alpha = 0.05$ |
| | Average | SD | Average | SD | Average | SD | | |
| silt + clay | 77.61 | 12.74 | 75.08 | 11.13 | 73.83 | 13.00 | 1.268 | 0.5306 |
| Density | 2.145 | 1.480 | 1.893 | 1.473 | 1.908 | 1.182 | 0.255 | 0.8805 |
| Hum. | 3.41 | 2.14 | 3.46 | 1.93 | 4.10 | 2.42 | 0.933 | 0.6271 |
| OC | 9.33 | 4.24 | 11.36 | 4.73 | 5.13 | 3.42 | 22.576 | $1.25 \times 10^{-5}$ *** |

**Table 2.** *Cont.*

| Variable | Use | | | | | | Kruskal (df = 2) | |
|---|---|---|---|---|---|---|---|---|
| | Abandoned N = 14 | | Cultivated N = 20 | | Industrial N = 33 | | Chi-Square | *p* Value |
| | Average | SD | Average | SD | Average | SD | | $\alpha = 0.05$ |
| TN | 2.28 | 0.77 | 3.25 | 1.09 | 1.83 | 0.89 | 22.921 | $1.05 \times 10^{-5}$ *** |
| C/N | 4.16 | 1.67 | 3.79 | 1.78 | 3.06 | 1.56 | 4.218 | 0.1213 |
| $CaCO_3$ | 360.44 | 51.60 | 346.46 | 65.59 | 362.20 | 78.46 | 1.396 | 0.4975 |
| pH | 8.55 | 0.34 | 8.55 | 0.27 | 8.32 | 0.38 | 6.998 | 0.03023 * |
| EC | 810.45 | 1537.17 | 507.57 | 468.48 | 1160.98 | 655.26 | 13.094 | 0.001434 ** |
| $P_{available}$ | 20.1 | 12.8 | 21.1 | 13.2 | 9.14 | 9.77 | 32.130 | $1.06 \times 10^{-4}$ *** |
| CEC | 14.581 | 4.062 | 14.723 | 3.254 | 13.248 | 3.875 | 3.765 | 0.1522 |
| $K_{available}$ | 36.055 | 11.998 | 35.164 | 24.306 | 24.392 | 12.271 | 7.774 | 0.02051 * |
| $Mg_{available}$ | 43.092 | 19.360 | 49.692 | 26.011 | 42.592 | 28.888 | 1.672 | 0.4335 |
| $Na_{available}$ | 26.444 | 23.672 | 29.950 | 31.115 | 65.808 | 210.310 | 0.057 | 0.972 |
| $Cu_{available}$ | 8.141 | 6.386 | 5.479 | 3.843 | 1.337 | 0.744 | 32.80 | $7.53 \times 10^{-8}$ *** |
| $Fe_{available}$ | 1.567 | 1.651 | 1.791 | 1.977 | 0.556 | 0.549 | 26.13 | $2.12 \times 10^{-6}$ *** |
| $Mn_{available}$ | 6.778 | 5.227 | 7.191 | 4.782 | 3.448 | 3.144 | 11.89 | 0.002621 ** |
| $Zn_{available}$ | 2.219 | 2.303 | 2.066 | 0.954 | 0.730 | 0.54 | 24.36 | $5.13 \times 10^{-6}$ *** |
| $Cu_{total}$ | 59.744 | 27.043 | 107.955 | 159.010 | 62.158 | 79.085 | 4.80 | 0.09056 |
| $Fe_{total}$ | 11,890.35 | 2596.26 | 11,289.95 | 2078.53 | 10,572.78 | 2069.75 | 4.01 | 0.1349 |
| $Mn_{total}$ | 212.850 | 38.037 | 231.050 | 41.663 | 211.985 | 39.692 | 4.40 | 0.1108 |
| $Zn_{total}$ | 36.996 | 28.817 | 52.378 | 36.805 | 23.266 | 21.516 | 12.82 | 0.00164 |

$p < 0.001$ ***; $p < 0.01$ **; $p < 0.05$ *.

**Table 3.** Statistics of the variables studied depending on the type of soil. Wilcoxon nonparametric test. Only those with significant differences are given.

| Variable | Wilcoxon Fluvisol-Regosol | |
|---|---|---|
| | W | *p* Value $\alpha = 0.05$ |
| OC | 206 | $8.69 \times 10^{-6}$ |
| TN | 920.5 | $6.71 \times 10^{-6}$ |
| EC | 272 | 0.0002967 |
| $P_{available}$ | 934 | $7.52 \times 10^{-9}$ |
| $Cu_{available}$ | 985 | $1.09 \times 10^{-7}$ |
| $Fe_{available}$ | 968.5 | $3.32 \times 10^{-7}$ |
| $Mn_{available}$ | 868 | $7.46 \times 10^{-5}$ |
| $Zn_{available}$ | 939 | $2.20 \times 10^{-6}$ |
| $Zn_{total}$ | 801 | 0.002282 |

Underlined $p < 0.05$.

**Table 4.** Statistics of the variables studied based on land use. Wilcoxon nonparametric test. Only those with significant differences are given.

| Variable | Wilcoxon Abandoned—Cultivated | | Wilcoxon Abandoned—Industrial | | Wilcoxon Cultivated—Industrial | |
|---|---|---|---|---|---|---|
| | W | $p$ Value $\alpha = 0.05$ | W | $p$ Value $\alpha = 0.05$ | W | $p$ Value $\alpha = 0.05$ |
| OC | 108 | 0.1664 | 398 | **0.0008** | 559 | **2.72 × 10$^{-8}$** |
| TN | 67.5 | **0.00625** | 34.5 | 0.0375 | 577 | **6.09 × 10$^{-6}$** |
| pH | 158 | 0.7897 | 333.5 | 0.05713 | 459.5 | 0.0179 |
| EC | 138 | 0.7138 | 122 | **0.004548** | 161 | **0.001563** |
| P$_{available}$ | 87 | 0.09786 | 409 | **8.18 × 10$^{-6}$** | 571 | **7.80 × 10$^{-8}$** |
| K$_{available}$ | 180 | 0.299 | 371 | 0.00622 | 415 | 0.121 |
| Cu$_{available}$ | 181 | 0.3136 | 427 | **0.00006843** | 617 | **1.46 × 10$^{-7}$** |
| Fe$_{available}$ | 119 | 0.3093 | 411.5 | **0.0002757** | 578 | **5.58 × 10$^{-6}$** |
| Mn$_{available}$ | 144 | 0.8564 | 362 | 0.01009 | 496 | **0.001918** |
| Zn$_{available}$ | 122 | 0.364 | 385 | **0.002309** | 586 | **2.75 × 10$^{-6}$** |
| Zn$_{total}$ | 102 | 0.1119 | 315 | 0.1258 | 520 | **0.0004065** |

Underlined $p < 0.05$; bold $p < 0.01$; underlined and bold $p < 0.001$.

### 3.1. Physical Properties of Soils

The physical characteristics of the soil are a necessary part of evaluating the quality of this resource because they cannot be easily improved [42]. The physical properties that can be used as indicators of soil quality are those that reflect how this resource accepts, retains, and transmits water to plants. Granulometry, real density and moisture content of these soils have been studied.

According to Gómez García [43], the predominant textures are loamy, with a clayey or silty tendency, but there are no statistically significant differences in the texture of the soils between Fluvisols and Regosols. They have an average sand content of around 25%, a little lower in the case of Fluvisols. The sandiest samples are found at the limits of the study area (Figure 3). Considering land uses, there are no statistically significant differences either, although industrial soils have a slightly higher sand content than cultivated soils and these are slightly higher than abandoned soils.

In 1992, a previous study [44] of the soils of the southern sector of Vega Alta del Segura already stated that they were fine-textured soils, predominantly silt loam, with slightly more fine sand in the C horizons, while the Ap horizons contain more clay. Our samples correspond to A horizons, which are Ap in many cases. We can say that the texture parameter does not allow us to discriminate between types of soils or their uses.

The soils studied do not present statistically significant differences in the real density parameter, neither between types of soils nor between their uses. The samples present a mean value of real density of 1.8–1.9 g/cm$^3$ (Table 1), which is a value that we can consider lower than the mean value for mineral soils given by [45,46] of 2.6 g/cm$^3$, perhaps due to the scarce clay content they have [46,47] and the absence of heavy elements. Regarding the water content, there are no statistically significant differences between the different types of soils or between their different uses. It can be said that physical properties do not allow differentiation of these soils, which aligns with the results obtained by [46,48].

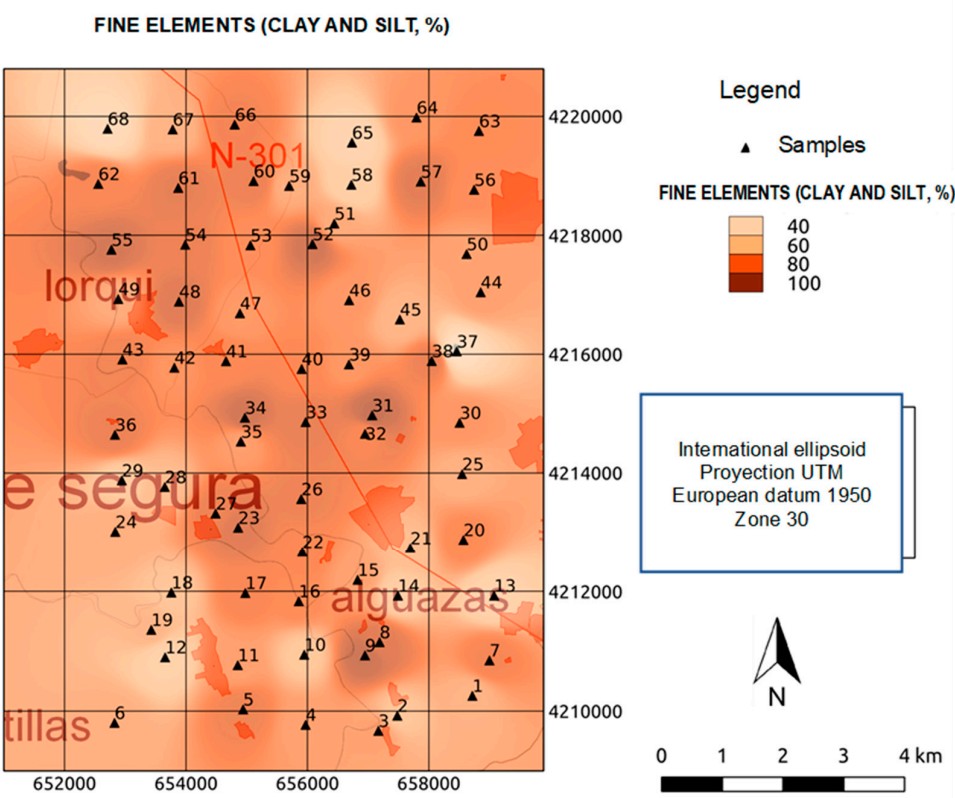

**Figure 3.** Interpolation of the silt and clay fractions in the study area.

### 3.2. Chemical Properties: Soil Fertility

The soils under study are very limestone. In fact they are calcaric Regosols and Fluvisols in both cases, with an average calcium carbonate content of more than 350 g kg⁻¹. They present a homogeneous spatial distribution in terms of carbonate content (Figure 4), not showing statistically significant differences regarding the type of soil or related to its use. The calcium carbonate content presents a statistically significant negative correlation (Table S6) with most of the assimilable micronutrients, justifiable by their immobilization at high pH. The average pH does not show statistical differences and corresponds to usual values in the Region of Murcia [49]. These values are in agreement with their high $CaCO_3$ contents.

Soil salinization is perhaps one of the most worrying problems for agriculture, and is responsible for its abandonment in many cases [50]. The presence of salts has consequences on the physical properties of the soil [51,52], on its toxicity and on the possible nutritional imbalance that may occur in the plants that settle in the soil [45,53]. The electrical conductivity values of the soil solution at a 1:5 ratio are higher in Regosols (1133.12 μS cm⁻¹), practically tripling those of Fluvisols (481.63 μS cm⁻¹), presenting very significant statistical differences between them ($p = 0.00029$) (Table 3) with a significance level of 0.05. Regarding use, very significant differences are observed between industrial use and abandoned ($p = 0.0045$) and cultivated ($p = 0.0015$) uses (Table 4), while there are no statistical differences between the currently abandoned or cultivated soils. Electrical conductivity value distribution is shown in Figure 5.

Once the EC was known, the ions of the extract were analyzed at 1:5, with a predominance of sulfates and chlorides among the anions, with nitrates being the minority. Among the cations, sodium and calcium predominate, followed by magnesium and potassium in a much smaller quantity.

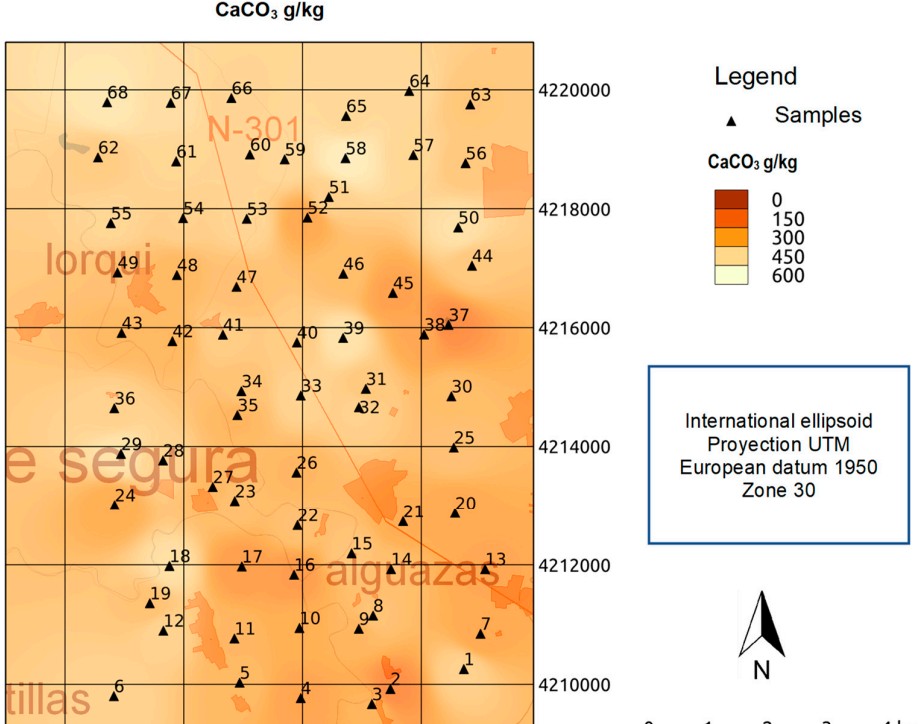

**Figure 4.** Interpolation of carbonates in the study area.

**Figure 5.** Interpolation of the EC in the study area.

Two of the areas monitored in the pilot action in the Region of Murcia, within the Desernet project [54], already revealed that problems of salinization and local contamination in the Segura River mid-plains are related to irrigation with low-quality water and urban industrial activities [55–57], although these studies stated that the Vega Alta del Segura had fewer salinization and contamination problems than the other monitored areas.

### 3.2.1. Organic Carbon

In respect of the organic carbon content, statistical treatment of the results obtained revealed the existence of very significant differences (at level 0.001) between Fluvisols and Regosols (with a *p* value of 3.65.10–2). Regarding use, the same statistical treatment also revealed statistically significant differences between industrial and agricultural use, with a *p* value of 1.25.10–5 at a significance level of 0.001 [43].

Highest organic carbon values are found in Fluvisols, which are located on both banks of the Segura River, with values of 15–20 g kg$^{-1}$, while lowest contents are found in Regosols, with values even lower than 5 g kg$^{-1}$; intermediate values are located spatially in the intermediate zone between them, with a clear gradation (Figure 6).

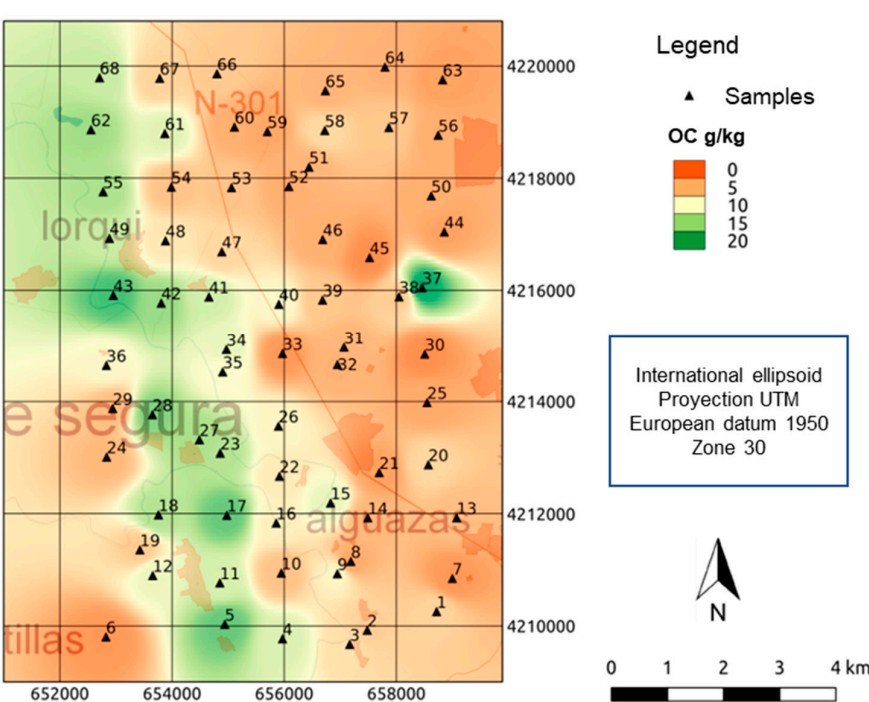

**Figure 6.** Interpolation of the OC in the study area.

Average values obtained in Fluvisols (10.58 g kg$^{-1}$) are very similar to those found by Marín in 1992, which were 9.4 g kg$^{-1}$ in the Ap horizons of Fluvisols sampled in the same area. This allows us to say that after 20 years, the organic carbon values have not decreased and that they are maintained over time. Values obtained by Carrasco (2000) when studying soils in the municipality of Torres de Cotillas are also along the same lines. Fluvisols present an abnormally low C/N ratio (mean value of 3.80), probably due to the fact that they have received nitrogen fertilization and to the fact that the mean value of total nitrogen is 2.91 g kg$^{-1}$.

In a similar study on a regional scale studying Fluvisols of Málaga, also in Spain [58], the intensification of human activities and inherent environmental conditions (e.g., topography, slope or climate) are changing the sustainability of fluvial soil ecosystems [59]. Variability of relationships between soil organic carbon and some soil properties in Mediterranean rangelands under different climatic conditions (South of Spain) was studied. Regosols, as

expected, present lower values, practically half (mean value of 5.26 g kg$^{-1}$). An exception occurs in sample 37 with a high OC value, despite being a gypsiferous calcaric Regosol, because it has a herbaceous cover that covers more than 40%. Regarding use, abandoned Fluvisols presented an average value of 9.33 g kg$^{-1}$ while those that are under a current crop have a higher average of 11.35 g kg$^{-1}$. Respecting industrial use, the average OC content is 5.13 g kg$^{-1}$. These results are aligned with those obtained in other abandoned farmlands in similar climatic conditions undergoing secondary succession processes [60–62] that showed little improvement or even decrease in soil carbon content after farmland abandonment because of the slow course of the soil under semiarid conditions.

### 3.2.2. Nutrients and Fertility

The values of exchange capacity in the study area can be classified as medium, since they are 14.69 mE/100 g for Fluvisols and 13.31 mE/100 g for Regosols (Table 1), without significant statistical differences between them ($p$ = 0.1572) or between uses.

Highest values occur in the soils located on both banks of the Segura River, mainly Fluvisols. Regarding the cation exchange capacity, it correlates positively and very significantly with the fine granulometric fractions clay and silt (0.539), since it is precisely the clay fraction that contributes to the formation of a stable humic clay complex responsible for the exchange capacity; however, organic carbon, the other fundamental constituent of this complex, also presents a positive but lower correlation (0.253) and at a lower significance level—$p < 0.05$ (Table S6). We can affirm that in these soils, the clay fraction contributes more to their exchange capacity than the organic matter.

The average value of total nitrogen in Fluvisols is 2.91 g kg$^{-1}$, while in Regosols it is 1.84 g kg$^{-1}$, presenting a very significant statistical difference between the soils (0.01425). Concerning the specific use of these soils, it should be noted that the currently cultivated Fluvisols present the highest average value (3.25 g kg$^{-1}$) with a distribution similar to that of organic carbon on both banks of the Segura River. Study [44] obtained lower values of total nitrogen (1.24 g kg$^{-1}$) in the Fluvisols studied in the same area, although they did not differentiate types of uses. The soils under study have a low organic matter content, although it correlates positively and very significantly with total nitrogen (0.461).

Carbon/nitrogen ratio in the studied soils is low (mean value less than 4, indicative of a good-quality humus mull), favored by the arid climatic conditions that rapidly mineralize the scarce organic matter.

The average content of assimilable magnesium does not present statistically significant differences between the different types of soils ($p$ = 0.0551). These are higher values than those found by [63] of 10.42 mg/100 g when studying the Segura River low plains and low Vinalopó river areas, but lower than those obtained by [44].

There are also no significant differences regarding the use of soils, as expected, being currently cultivated soils, with the highest value of assimilable magnesium (49.69 mg/100 g).

The average content of assimilable potassium presents low statistically significant differences ($p$ = 0.0205) with respect to the use of soils; however, there are no statistically significant differences between the two types of soils. Intermediate values (29 mg/100 g) were obtained by Marín (1992) in Fluvisols, studying potassium fertility of the soils of the southern sector of Vega Alta del Segura.

Assimilable phosphorus contents are low or very low, according to the high or very high values of total and active calcium carbonate [64]. Highest values of assimilable phosphorus correspond to Fluvisols (23.89 mg·kg$^{-1}$), a mean value similar to that of 24.41 mg·kg$^{-1}$ obtained by [65] in other soils in the region of Murcia.

The statistical treatment of the results obtained has revealed the existence of significative differences between Fluvisols and Regosols, as well as between industrial and agricultural use, in terms of the content of organic carbon, total nitrogen and assimilable phosphorus. Higher values were found in calcaric Fluvisols, as expected from other studies [43,66]. Regarding use, there are very significant statistical differences between agricultural use and industrial use; these significant differences only appear between cur-

rent and past agricultural use in total nitrogen and assimilable potassium content (Table 4), which is in line with the results obtained by many other authors. (i.e., [66–69]).

In general, comparing the different types of soils, there are no statistically significant differences in the total contents of any micronutrient (with the exception of zinc), while there are differences in their assimilable contents ($p < 0.001$), presenting the highest values in Fluvisols, in accordance with the fact that they are soils with a clear agricultural vocation. However, when comparing those currently cultivated with those that are not cultivated, there are hardly any differences. The rest of the elements (copper, iron and manganese) present very significant differences in their assimilable content, but do not present significant differences in their total contents when studying the different types of soils.

Sampled calcaric Fluvisols present an average value of total copper of 89.62 mg·kg$^{-1}$ and calcaric Regosols 62.09 mg·kg$^{-1}$, with no significant differences between them (Table 3). However, in terms of assimilable copper, there are very significant statistical differences ($p = 0.000473$) between the soils. In the case of calcaric Fluvisols, the assimilability index is higher and 10% of the total copper is in assimilable form, while in the case of Regosols only 2.8% is available. All the values are included in the wide range given by [54] for the Region of Murcia. Regarding use, soils located in industrial locations present the lowest values of assimilable copper (1.33 mg·kg$^{-1}$), with very significant statistical differences with the cultivated soils, which may be attributed to soil physicochemical properties and/or by the rhizosphere environment, leading to a release of (bio)available metals [70]. The highest values of assimilable copper are found in samples 62, 66, 24 and 6, all of them abandoned calcaric Fluvisols, which were cultivated in the past.

Assimilable iron values of the sampled soils can be classified as very low according to the classification established by Lindsay and Norvell in 1969, which is still valid according to [71], as they are less than 2 mg·kg$^{-1}$, even in currently cultivated soils with a value average of 1.79 mg·kg$^{-1}$, showing very significant statistical differences between cultivated soils and those dedicated to industrial use ($p = 0.0000$). These assimilable iron values are similar to those obtained by [44] in the Ap horizons of his sampled profiles (1.65 mg·kg$^{-1}$).

The iron assimilability index can be said to be practically null, since from the average values of total iron (11.60 mg·kg$^{-1}$ in Fluvisols and 10.56 mg·kg$^{-1}$ in Regosols), the assimilable part is less than 0.01%, justifiable by the high pH of these limestone soils that causes practically all of this element to be immobilized in the form of precipitated and insoluble oxides and hydroxides. This low availability of iron is responsible for the iron chlorosis that affects plants. Therefore, iron chlorosis manifests itself with yellowing of the leaves, which was also observed in many fruit trees grown in the study area. Due to this problem with the immobility of iron, it is fertilized with chelates that keep it in solution. On the other hand, the role of organic matter is also very positive, since the formation of chelates between organic matter and iron favors its transition to available forms [72].

Manganese is the element with the lowest assimilability index after iron; therefore, we can affirm that practically all the manganese in the soil is found in an unavailable form due to being very limestone soils, with high pH and little organic matter. The values of assimilable manganese present significant differences between the different types of soils ($p = 0.00018$) and between the uses to which they are dedicated ($p = 0.00262$). The mean values of assimilable manganese (7.34 mg·kg$^{-1}$) are higher than those obtained by [44] (2.04 mg·kg$^{-1}$ for Ap horizons). The average values of total manganese in calcaric Fluvisols (224.51 mg·kg$^{-1}$) are very similar to those obtained by this same researcher, of 248 mg·kg−1 for the Ap horizons and 202.5 mg·kg$^{-1}$ for the C horizons.

Assimilable zinc content presents very significant differences between the two types of soils studied ($p = 0.004254$), and it is the only one of the micronutrients analyzed that also presents significant statistical differences ($p = 0.002572$) in its total content for the types of soils. In calcaric Fluvisols, the average value of total zinc (44.16 mg·kg$^{-1}$) is very similar to that found by [73] in similar areas and practically double the average value of total zinc in calcaric Regosols (23.75 mg·kg$^{-1}$). The assimilable zinc contents are similar in all Fluvisols,

both cultivated and abandoned, slightly higher than 2 mg·kg$^{-1}$ and very similar to those found by Marín [44] in their studied Fluvisols (1.52 mg·kg$^{-1}$). The assimilability index is therefore situated at 13%, coherent with the calcareous nature of these soils [70,74]. Soils under industrial use present lower values (0.73 mg·kg$^{-1}$).

It is advisable to apply fertilizers with a low saline index and use adequate doses that do not increase the salinity of the soil. It is also recommended that the traditional irrigation and eco-structures be maintained, e.g., the maintenance of hedgerows, ditches, and high-value habitats such as ponds, due to their contribution to maintain water flows through alluvial aquifers as well as the biodiversity through the species typical of riparian environments and wetlands; therefore, may also be considered nature based solutions (NbS) insofar as they constitute green–blue corridors that also provide leisure spaces and constitute a cultural heritage, all in a sustainable manner. Another interesting option to consider would be to use crop varieties with genetic improvements more tolerant to salts. It is very important to integrate the traditional garden, a special sign of identity, into the urban landscape.

## 4. Conclusions

In view of the results obtained, it can be said that physical properties do not allow differentiation of these soils. The statistical treatment of the results obtained revealed the existence of significative differences between Fluvisols and Regosols, as well as between industrial and agricultural use, for the content of organic carbon, total nitrogen and assimilable phosphorus. Regarding the spatial distribution of organic carbon in these soils, it can be concluded that although it presents some irregularity, it depends fundamentally on the type and use of the soil and the human influence on them. Higher values were found in calcaric Fluvisols.

In general, comparing the different types of soils, there are no statistically significant differences in the total contents of any micronutrient (with the exception of zinc), while there are differences in their assimilable contents, presenting the highest values in Fluvisols.

According to the starting hypothesis, the soils are dedicated to agricultural or industrial use according to their intrinsic characteristics: preferably the Regosols are in industrial use and the Fluvisols under agricultural use.

The study presents a sustainable and resilient agropolitan system in which different types of soils and uses are presented. During the field trips, an abandonment of the cultivated fields has been verified, especially in the agricultural areas of low economic profitability, due to a change in the tasks of the inhabitants of the area that nowadays are more dedicated to industrial and service activities. All of this has been accelerated by the aridity of the climate and demographic pressure. It may be concluded that adequate agricultural practices that do not abuse fertilization or the use of irrigation water, mostly coming from the Segura River, can be considered to be of acceptable quality. The agricultural soils have maintained current and potential fertility and do not suffer from salinization problems.

**Supplementary Materials:** The following supporting information can be downloaded at https://www.mdpi.com/article/10.3390/f14061085/s1. Table S1: Type, use and coordinates of soil samples; Table S2: General analysis of soil samples; Table S3: Granulometric analysis of soil samples; Table S4: Exchange capacity and bases of exchange in soil samples; Table S5: Available and total metals; and Table S6: Correlations between the variables studied.

**Author Contributions:** Conceptualization, P.M.-S. and M.A.A.-L.; methodology, P.M.-S. and J.M.G.-V.; software, A.B.-B.; validation, M.A.A.-L., P.M.-S. and A.B.-B.; formal analysis, A.M.G.-G. and J.M.G.-V.; investigation, P.M.-S. and M.A.A.-L.; resources, P.M.-S.; data curation, A.M.G.-G. and J.M.G.-V.; writing—original draft preparation, P.M.-S., M.A.A.-L. and A.B.-B.; writing—review and editing, P.M.-S., M.A.A.-L. and A.B.-B.; visualization, A.B.-B.; supervision, all authors. All authors have read and agreed to the published version of the manuscript.

**Funding:** This research received no external funding.

**Institutional Review Board Statement:** Not applicable.

**Informed Consent Statement:** Not applicable.

**Data Availability Statement:** The data presented in this study are available in Supplementary Materials.

**Conflicts of Interest:** The authors declare no conflict of interest.

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
