# Peer review of "Influence of the Type and Use of Soil on the Distribution of Organic Carbon and Other Soil Properties in a Sustainable and Resilient Agropolitan System"

_forests, doi:10.3390/f14061085_

Round 1

Reviewer 1 Report

This research manuscript entitled “Influence of the type and use of soil on the distribution of organic carbon and other soil properties in a sustainable and resilient agropolitan system” investigated the soil properties in the peri urban area in the Vega Media of Segura River and discussed the differences in soil properties among soil type or land use. While I appreciate the effort of this work, I think this manuscript needs some revisions.

I suggest authors reconsider the data analysis method. It causes confusion to include the data on Solonchak for statistical analysis although authors describe that the presence in the study area of a single sample of Solonchak has not been considered representative and therefore not included in the results and discussion (Line 208–209). It is important to show the observed value of Solonchak, however I doubt whether just one sample value of Solonchak can be significantly compared with those of other soil types. Additionally, it is necessary to discuss the combined effect of both soil type and land use management on soil properties by statistical analysis, such as two-way Anova or Friedman test although authors analyzed each effect on soil properties separately. The result of this analysis should lead to the finding to utilize and develop the land effectively and sustainably based on relatively stable condition of each soil type.

Then, authors should carefully review and correct this manuscript because it contains spelling inconsistencies and grammatical mistakes as research articles in English, such as inconsistent description of statistical results, inconsistent units, and subject omission.

I hope these comments would be helpful.

Author Response

Dear reviewer we attach our answers to your comments.

Thank you very much as they will for sure improve our paper.

Best regards. 

Reviewer 2 Report

See attched file

Author Response

Dear reviewer

We attach our answers to your comments.

Thank you very much as they will for sure improve our paper.

Best regards. 

Reviewer 3 Report

Review of the manuscript: Influence of the type and use of soil on the distribution of organic carbon and other soil properties in a sustainable and resilient agropolitan system

General comments: The manuscript is not well written in English and reports the assessment of the distribution of organic carbon and to study other indicators of soil quality in the “Huerta de Murcia” in Spain. The introduction and discussion are well documented. However, there are some specific comments for the abstract and conclusions section that should be resolved before publication.

Comments to authors:

Typographical errors: In lines 34, 60, and 69, unnecessary periods and/or commas should be deleted. Likewise, in lines 52, 63, and 70, a period is missing at the end.

In line 155, the word "various" is an error.

In lines 318, 319, and 325, it says “Kg-1” and it should say "Kg-1".

In figure 1 (line 157), geographic coordinates should be included on the map.

The abstract needs improvement, starting with a brief description of the importance of the research topic, followed by the research objective, methods, results, and conclusions.

The objectives of the work are not clear and do not correspond to the specific research topic. Additionally, the hypotheses of the work are not clearly defined and should be appropriately mentioned in the manuscript at the end of the introduction section.

Finally, the conclusions are not based on the results or do not directly correspond to the specific research topic investigated and should be improved by providing an effective synthesis of the results and implications of the research. It is important to highlight the most significant findings and refer to the hypotheses (not stated at the beginning of the paper), emphasizing the conclusion for each specific objective set in the work and highlighting the important contributions of the research, with a reflection on the relevance and importance of the research and its contribution to the field of study.

Author Response

(The authors gave the same response as above.)

Round 2

Reviewer 1 Report

I appreciate authors’ thoughtful response to my peer review comments. I have confirmed that they were approximately addressed, and the manuscript was improved well. On a minor point, please consider the following specific points again.

All units should be described according to SI Units. For example, the unit of density should be g cm-3 or Mg m-3, not g/cc. Additionally, I think it is better to unify the notation of units in both tables and the main text (e.g. EC) for readers to understand this study paper easily. Then, I think the P value should be rounded off to third decimal place and described as ‘P < 0.001’ if P value is less than 0.001, because describing such small value accurately is not very important.

I hope these comments will be helpful.

Author Response

We appreciate reviewer`s comments as they will improve our paper.

We have unified the notation of units, according to SI, in both tables and the main text, and also in supplementary material. 

We think that it is important to keep "p" values with all decimals, as it gives more information.

Thank you very much for your valuable help.

Best regards.